# Human-in-Loop Decision-Making and Autonomy: Lessons Learnt from the Aviation Industry Transferred to Cyber-Physical Systems

Chara Makri [1,*], Didem Gürdür Broo [2,*] and Andy Neely [3]

1 Centre for Digital Built Britain, University of Cambridge, Cambridge CB3 0FA, UK
2 Center for Design Research, Stanford University, Stanford, CA 94305, USA
3 Institute for Manufacturing, Department of Engineering, University of Cambridge, Cambridge CB3 0FS, UK
* Correspondence: cam221@cantab.ac.uk (C.M.); didem@stanford.edu (D.G.B.)

**Abstract:** In this study, we reviewed aircraft accidents in order to understand how autonomy and safety has been managed in the aviation industry, with the aim of transferring our findings to autonomous cyber-physical systems (CPSs) in general. Through the qualitative analysis of 26 reports of aircraft accidents that took place from 2016 to 2022, we identified the most common contributing factors and the actors involved in aircraft accidents. We found that accidents were rarely the result of a single event or actor, with the most common contributing factor being non-adherence to standard operating procedures (SOPs). Considering that the aviation industry has had decades to perfect their SOPs, it is important for CPSs not only to consider the actors and causes that may contribute to safety-related issues, but also to consider well-defined reporting practices, as well as the different levels of mechanisms checked by diverse stakeholders, in order to minimise the cascading nature of such events to improve safety. In addition to proposing a new definition of safety, in this study we suggest reviewing high-reliability organisations to offer further insights as part of future research on CPS safety.

**Keywords:** cyber-physical systems; human-in-loop decision-making; autonomy; accountability; through-life accountability; safety

## 1. Introduction

The word "autonomy" originated from the Greek word "autonomia/autonomos", the meaning of which can be expressed as "auto = self" and "nomos = law" [1]. It can also be defined as independent and self-governing [2]. Though the term autonomy and related concepts exist in various kinds of sciences, the engineering etymology is concerned with "*the ability of an engineering system to make its own decisions about its actions while performing different tasks, without the need for the involvement of an exogenous system or operator*" [3]. Furthermore, the word "automation" was inspired by an earlier term, that is, "automatic". Automation includes the execution by a machine agent of a function that was previously carried out by a human [4,5]. Even though the two terms seem connected and are sometimes used interchangeably, they have different meanings. Automation refers to a system that will do exactly what it is programmed to do by the programmer, without having any choice or possibility to act in a different way depending on the situation at hand. The main difference between autonomy and automation, or autonomous and automated systems, is related to their ability to change their actions in the future. In automated systems, actions are predefined from the beginning, and they have no ability to change their actions in the future.

Cyber-physical systems (CPS) are defined as systems integrating physical processes, computations, and networks through feedback loops in which physical processes affect computations and vice versa [6]. They represent the collaborative robots, autonomous

vehicles, smart factories, and cities of today and tomorrow [7–9]. These systems are not only smart/intelligent but are also highly automated and they are expected to act autonomously in the future. However, the level of the autonomy of these systems is still relatively low and, at least for the near future, their operations are not expected to be fully autonomous. Instead, there will be human–machine collaborations, automation, shared autonomy, and human-in-the-loop decision making. At present, we are already experiencing difficulties related to the transitioning from human-controlled to human-in-the-loop systems and to autonomous systems. Some examples of these difficulties and challenges have been studied by researchers [10–12], who investigated the number of traffic accidents that occurred with autonomous vehicles and then compared these data with the data concerning traffic accidents involving conventional vehicles.

At the same time, modern aircraft systems are seen as one of the most automated systems currently available. They can augment pilot performance or even replace pilots with the same performance level in tasks such as managing engine power, controlling and navigating the aircraft, and in some cases even completing landings. Although the aviation industry and the systems it uses have undergone several technological transformations in the areas of communication, navigation, surveillance, guidance, and control-related aspects of automation and human-in-the-loop decision making [13], in many other industries and sectors these transformations are still ongoing. The differences between different industries may be seen as an opportunity to help us learn from one industry and not only use that knowledge to advance others, but also avoid repeating the same mistakes when adopting new technologies and driving transformation.

In this study, we aimed to critically review aircraft accidents in Supplementary Materials as a way to understand how autonomy and safety has been managed in the aviation industry and to transfer our findings to other venues in which autonomous CPSs are going to be implemented. Since we aimed to translate our findings from the aerospace industry and generalise them as a set of guidelines for the design of autonomous cyber-physical systems, the main purposes of this study were expressed in the form of the following research questions:

- Main research question (RQ): What are the important considerations in the design of future autonomous and intelligent systems (CPSs) when focusing on the results regarding the safety of decision-making processes and the accountability of these decisions?
- Sub-RQ1: What can we learn from the aviation industry in regard to human-in-the-loop decision-making by reviewing accident reports and considering accountability and safety as the main two metrics?
- Sub-RQ2: How can these findings then be translated to developing a methodological approach to be used in the design of autonomous CPSs?

To this end, we first undertook a critical review of human-in-the-loop decision-making and its limitations by conducting a review of the literature on human-in-the-loop decision making, accountability, and safety. Secondly, we reviewed a series of aircraft accident reports and highlighted the roles of different stakeholders and their involvement in the accidents. To clarify the roles of stakeholders and their involvement, the concept of accountability played a significant role in this study; therefore, the relevant earlier work on accountability, its relationship with safety, and the reasoning behind using accident reports for gathering knowledge are explained in detail.

The rest of the paper is organised as follows: Section 2 presents the extensive literature review on human-in-the-loop decision making, accountability, and safety. Section 3 describes the methodology that was adopted throughout this study. Section 4 presents the data analysis and the results of the study. Section 5 presents a detailed discussion on our findings and suggests ways to transfer knowledge between industries. The final section presents our conclusions and future research directions.

## 2. Literature Review

In this section, we present relevant literature on human-in-loop decision-making, accountability, and safety, with the aim of highlighting the previous studies conducted on these topics, to show their limitations and relevance with regard to the focus of the present study and to inform the reader about how this study relates to these important research topics.

### 2.1. Human-in-the-Loop Decision-Making

Through CPSs, many autonomous capabilities have been gradually introduced to the automotive, manufacturing, and robotics industries, as well as many other industries [14]. However, there are still a lot of tasks in which human participation is required in order to accomplish successful operations. This human participation is called human-in loop decision-making. In this way, human-in-the-loop decision-making combines the cognitive skills of humans with the behaviours of autonomous systems. In other words, human-in-loop systems require human support to guarantee their complete and correct behaviour in all situations [15]. In this section, we present some of the most relevant research on human-in-the-loop decision-making, with a focus on studies related to CPSs.

Many researchers have contributed to the literature on human-in-the-loop decision making. Some have focused on control-related aspects of these decisions. For instance, Feng et al. [16] proposed an approach to synthesising control protocols for autonomous systems that accounts for uncertainties and imperfections in interactions with human operators. They used a case study in which road network surveillance was performed by an unmanned aerial vehicle, and the remote controlling of the vehicle involved a human operator and a certain degree of autonomy. In addition, other studies have focused on human factors in human-in-the-loop decision-making. Jirgl et al. [15], for instance, presented different approaches and provided an example of human performance assessments regarding the prediction of probable human responses and their dynamic properties. They obtained data from flight simulators during interaction with real pilots and the results indicated that there was a possibility of predicting a probable human behaviour based on their measurements and modelling approaches.

Sousa Nunes et al. [17], on the other hand, adopted a critical view of the natural confluence of the multidisciplinary focus on human-in-the-loop CPSs, stating that the current practices lack a general understanding of the underlying requirements, principles, and theory. They discussed the current state of the art of human-in-the-loop CPSs and provided a critical overview of the current taxonomies. Their findings highlight that there are several technical and ethical limitations that have yet to be completely resolved in the current research. Similarly, Gil et al. [14] also focused on CPSs and identified the technological challenges involved in designing human-in-the-loop CPSs. Their approach suggested that designers should identify and specify how humans and systems should work together, while focusing on control strategies and interactions. Emmanouilidis et al. [18] built on this idea, describing emerging technologies as enablers to empower human operators to become more effectively integrated in production activities [19] and presenting a viewpoint about the enabling of human-in-the-loop engagement linked to cognitive capabilities in industrial CPS.

### 2.2. Accountability

Another important body of work relates to the concept of accountability. There are several reasons why accountability matters. First, a thorough understanding of accountability for a human-in-the-loop CPS provides better control over the system. Second, such an understanding can shield the organisations involved against liabilities (including reputational damage) that might occur as a result of a failure. Third, and perhaps most importantly, an ex-ante understanding of the sources of liabilities following the failure of a human-in-the-loop CPS can provide key insights for improving the safety of the system. A single search of the term in online research databases, however, returned more than 50,000 results, dating

back to the 1970s. Given the long history of the term, it is to be expected that the relevant literature is voluminous. Nevertheless, most studies have focused on identifying the exact meaning of the term and exploring its different forms in the context of accounting, political science, and sociology. While a systematic literature review can minimise bias and provide high-quality research, in fields that consist of many autonomous subfields, it can lead to high volumes of information and a lack of "*transdisciplinary understanding*" [20]. Thus, for the purpose of this study, a broader review of the literature was conducted with the search focusing on gaining a better understanding of its meaning and identifying the main issues discussed in the literature.

The term accountability has its roots in ancient Egypt and Greece [21,22], and comes from the Latin word "accomptare", which means "to give account" [23]. Although accountability is also often used as a synonym of the terms responsibility and liability, its meaning has been extended beyond these simple terms, in multiple directions, resulting in authors often calling it a chameleon-like term [24,25].

Two distinct streams were identified in the literature. In the first, authors argued that accountability has different forms and attempted to identify them. In more detail, accountability can be differentiated into individualising (internal or personal) and socialising (external or structural) forms [24–28]. According to these studies, accountability can be further broken down into hierarchical and professional forms, representing personal accountabilities, and into legal, public, and political forms, corresponding to external accountabilities. An overview of the different forms can be found in Figure 1.

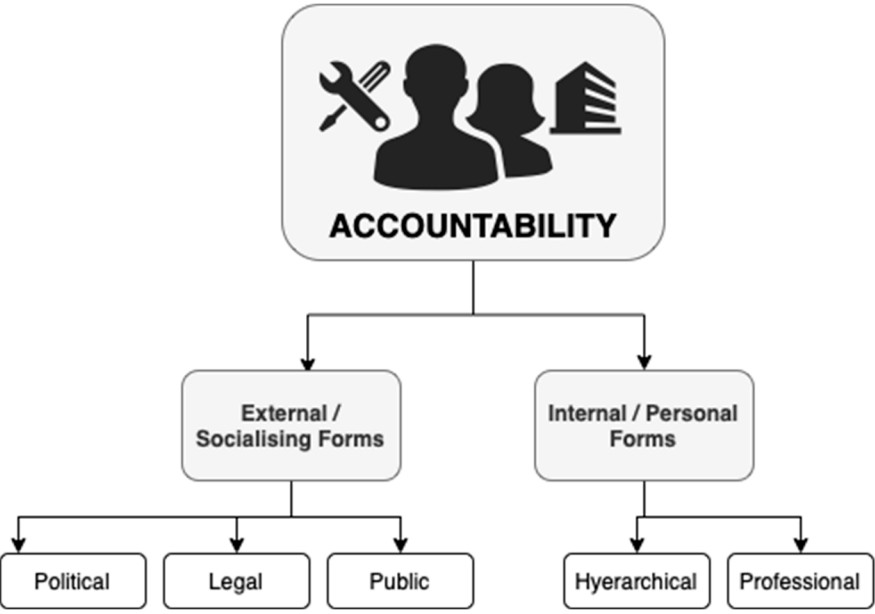

**Figure 1.** Different forms of accountability.

For organisations, firms, and society as a whole to secure prosperity, they need to find a balance between the forms of accountability and align the pursuit of their economic goals with a wider "*ethical discourse*" [28–31]. However, finding the balance between the different forms of accountability can only be achieved under ideal conditions [24,32]. This leads to the second stream of the literature, in which researchers argue that the multiple forms of accountability can lead to conflicting situations—*multiple accountabilities disorder* (MAD) [33]—since actors and organisations may be accountable for more than one dimension [24,34]. Roberts [26], for instance, explains that accountability acknowledges that one's actions make a difference to others, and argues that individualising forms of accountability produce a "*solitary*" sense of oneself. These individualising forms come into contradiction with the socialising forms of accountability, "*which confirm self in a way that emphasizes the interdependence of self and others*". Behn [35] states that in accountability there

are two different types of actors: the holder and the "*holdee*" (see also [36]), with the latter being called to account for both objective targets and subjective ones. The author argues that in contrast to objective targets, subjective ones are particularly difficult to quantify. Therefore, more attention is given to the former, resulting in an "*accountability bias*" [35].

The complex meaning of accountability and its multiple forms can therefore lead to conflicting situations, since actors are often required to give account to a variety of different stakeholders against a broad range of criteria. Thus, it is far from straightforward for organisations to understand and deal with their accountabilities, especially in organisations that operate in large networks of partners. Bovens and Hix [37], for example, explain that in order to identify the forms of accountability, one needs to answer four questions: accountability to whom, who will be held accountable, accountable about what, and why the person accountable has the need to give account. When trying to answer the first question, organisations are often faced with "*the problem of many eyes*", and when trying to answer the second question they are often faced with "*the problem of many hands*". Nonetheless, although the problem of many eyes has been extensively discussed in the literature, the problem of many hands has mainly been examined in terms of the setup of different actors within the same organisation. Therefore, how conflicting accountabilities can be handled when multiple organisations are involved is an issue that has still to be addressed. Moreover, in cases of failure, dealing with accountabilities can become even more complex. Romzek and Ingraham [27], for instance, analyzed the so-called Ron Brown plane crash. On April 3, 1996, a USAF Boeing CT-43A crashed on a mountainside during its approach to Dubrovnik, Croatia, killing everyone on board. According to the official report the crash was the result of several factors: "*failure of command, aircrew error and an improperly designed instrument approach procedure*", as referenced by the Flight Safety Foundation – Flight Safety Digest, [38]. Romzek and Ingraham used this case study to explain how accountability becomes a challenge when things go wrong and conflicts arise among the various elements of accountability [38] (p. 1). In particular, they demonstrated that although professional forms of accountability require individuals to take initiative to complete their mission, the crew involved in the crash were caught between conflicting rules. Nonetheless, they were held accountable, even though their actions may have been the most reasonable ones to take at the time of the crisis.

Another concept that falls under the body of literature on accountability is the concept of through-life accountability (TLA). This concept was first introduced by Fuse [39] and was defined as " . . . *the duty to inform, justify and accept the consequences of decisions and actions taken during the entire lifecycle of assets and associated services. Critically it involves understanding the boundaries of and responsibilities for safe and consistent outcome delivery over an extended service contract involving multiple organisations*". Fielder et al. [40] emphasised the importance of TLA by describing the business model of Her Majesty's Naval Base (HMNB) Portsmouth, an example taken from BAE Systems Maritime Services, in which the company had a contract to maintain maritime vessels. As part of their role, BAE had also been tasked with maintaining and running the Naval Base. This included providing catering, accommodation, and social facilities for many employees, including sailors. Since BAE did not have expertise in all these areas (e.g., catering) they outsourced parts of the contract to external contractors, but the company was still responsible in cases in which the agreed-upon standards were not met by any of the partners. As the authors pointed out, "*understanding accountability in this context therefore becomes essential if we are to manage the associated risks and liabilities*".

A thorough understanding of TLA, therefore, provides organisations operating in complex systems with better control over their networks. Furthermore, such an understanding can shield them against potential risks. Interestingly, however, the relevant academic research on TLA is sparse at best. For example, using "*Through-Life Accountability*" as the search term in online research databases (e.g., Scopus® and ScienceDirect®) returned zero results. The same search in Google Scholar® only returned the abovementioned study by Fielder et al. [40], whereas a Google® search returned just over 1000 results, almost none

of which were relevant to the term, apart from the ones specifically mentioning the study by Fielder et al. [40], and without offering any further investigation or insights (based on searches performed in October 2014, when this study started). One of these search results mentioned the term and its importance, but without further evidence or definitions [41]. A later search revealed one more result mentioning the term and its importance, but again without any further discussion or evidence [42]. The remaining results mostly referred to one's personal accountability.

Although numerous studies have discussed the complexity of accountability, there is a paucity of research on guidance or tools that can be used by organisations for identifying and managing accountabilities. In this direction, Roberts [43] suggested that societies and organisations have a need for a new intelligent form of accountability. This should not just be a snapshot of performance, but should also be interactive, evolve over time through active dialog and communication, and have the ability to test outcomes and provide improvements, according to both moral and strategic criteria [44].

*2.3. Safety*

Safety is a term that is both widely used and highly recognizable. The results of a search for the word "*safety*" prove this point; in Google Scholar®, the search returned more than 3,000,000 results, whereas a search in the Scopus® database returned more than 1,000,000 results (based on searches performed in November 2017). Nevertheless, because everyone has an intuitive understanding of the term safety, few efforts have been made to clearly define it [45]. Indeed, by using the term "*safety definition*" and looking only at peer-reviewed papers (i.e., limiting our search to "*articles or reviews*") the search results in Scopus® were reduced to 29. Of these, 15 did not provide a definition of safety, 7 referred to patient safety, 2 were written in German, 2 were not available, 2 recommended technical definitions that enabled numerical calculations of the number of faults in technical systems, and only 1 provided a qualitative definition of safety. According to that study, safety is a state in which there is no "*unacceptable risk*", and may also be defined as the "*antonym of risk*" [46]. In another example, the Oxford Living Dictionaries [47] defined safety as "*the condition of being protected from or unlikely to cause danger, risk or injury*", whereas, according to the *Business Dictionary* [48], safety is "*relative freedom from danger, risk or threat of harm, injury, or loss to personnel and/or property, whether caused deliberately or by accident.*" Different versions of these definitions can also be found in industry reports in sectors in which maintaining a high level of safety is critical, such as the aviation and nuclear sectors. For example, the International Civil Aviation Organization (ICAO) defines safety as "*the state in which the possibility of harm to persons or of property damage is reduced to, and maintained at or below, an acceptable level through a continuing process of hazard identification and safety risk management*" [49] (para. 2.1), whereas the International Atomic Energy Agency (IAEA) defines (nuclear) safety as "*the achievement of proper operating conditions, prevention of accidents or mitigation of accident consequences, resulting in protection of workers, the public and the environment from undue radiation hazards*" [50].

Therefore, it is clear that the definitions of safety largely rely on the understanding of the relevant risks, or as Amundrud et al. [46] explained, "*the safety level is linked to the risk level*" and "*a high safety means a low risk and vice versa*". Moreover, the use of words such as "*unlikely*" and "*acceptable level*" reveal that identifying the acceptable level of safety, and therefore of risk for each organisation, is far from straightforward. This situation has led several scholars to argue that safety is not only complex, but also relative [51–53]. Especially in cases of organisations that operate in complex networks, identifying the risks and the actions necessary to manage them can be especially challenging.

Furthermore, Hollnagel et al. [45] pointed out that the complex systems in modern economies have made the way that we understand and deal with safety redundant. They recommend a shift from earlier approaches to safety, which they define as Safety I, towards a new approach that they call Safety II. According to their report, Safety I is defined as "*a state where as few things as possible go wrong*" and attribute the causes of failures to technical,

organisational, or human factors. Instead, they define Safety II as "*a state where as many things as possible go right*" [45]. Both definitions, however, present the same challenges as the ones discussed previously.

### 3. Methodology

In this section, in addition to explaining the overall methodology that was followed throughout this study, we aimed to present the reasoning behind our focus on aircraft accident reports and to explain how a deep understanding of the causes of these accidents may help promote safety.

The methodology that was followed in this study started with identifying important commonalities between the highly automated systems used in the aviation industry and CPSs. To this end, we reviewed the literature related to human-in-loop decision-making, accountability, and safety. The results of these investigations are presented in Section 2 in order to help the reader understand the important backgrounds of these concepts, and how they are related to future CPSs and current aviation industry practices.

The United Nations' International Civil Aviation Organization defines an aircraft accident as "*An occurrence associated with the operation of an aircraft which takes place between the time any person boards the aircraft with the intention of flight until such time as all such persons have disembarked...*" in which "*a person is fatally or seriously injured*", "*the aircraft sustains damage or structural failure*", or "*the aircraft is missing or is completely inaccessible*" [54]. In the case of such an occurrence, a formal investigation takes place with the purpose of gathering and analyzing all available data in order to uncover the causes of the accident. The conclusions of the investigation are summarised in a final report that includes a detailed timeline of the accident, all relevant evidence gathered, and an assessment of the causes of the accident, and this report may also include safety recommendations. To this end, aircraft accidents provide an ideal resource for the purposes of this study.

Using the information provided in official aircraft accident reports, we performed both conventional and direct content analysis [55,56] with the use of NVivo® software [57]. NVivo® is a computer-assisted qualitative data analysis software (CAQDAS). The use of CAQDAS is recommended in qualitative analysis, as it can support the analysis in various ways [58]. For example, according to the authors, using a CAQSAS can support researchers when using different types of data (e.g., qualitative and quantitative) and can also make the relationships between the nodes more clear and easier to verify. In our case, the use of software such as NVivo® was deemed necessary not only due to the large number of analyzed incident reports but also due to the fact that these reports were significantly lengthy. There are several studies in the literature that provide detailed presentations of how to use CAQDAS, including NVivo®, in qualitative research (see, for example, [56]). It should be noted, however, that the use of software such as NVivo® only provides support to the researcher and does not perform the analysis for them [58].

In the case of this study, we began the analysis by reading each accident report carefully, focusing specifically on the 'conclusions' sections, where the causes of each accident were summarised in detail. During this process, we highlighted any text that described a cause for the accident and assigned it to a 'node' describing the specific cause, for example, 'lack of proper maintenance'. Each node was also assigned to a 'case' connecting each cause to an actor (see also, [59]). For example, the 'flight crew' comprises the pilot, co-pilot, and flight attendants, and the approach controller and air-traffic controller form the 'air-traffic control' group. The groups identified in this stage were then used to address RQ2.

Any reference to an accident's cause, or an actor, that was not clear was assigned a separate general code, in order that it could be reviewed again at the end of the process and approved by another researcher. Once all the selected accidents were coded, we reviewed all the nodes again and analyzed their relationships. As a result of this process, some nodes were merged into bigger 'parent nodes', whereas others were merged under the wider 'parent' nodes as 'child' nodes. Further analysis of these characteristics was then used to help us gain knowledge on how the findings from the aviation industry could be translated

to the field of CPS design. For example, by compiling the list of different accident causes, we were able to understand the most common types of accident causes and provide safety recommendations, which could be translated into a CPS design.

## 4. Data Analysis and Results

In this section, we present a detailed explanation of the data collection process, as well as the results obtained in the preliminary analyses and in the detailed analyses of accident reports.

### 4.1. Data Collection

First, we identified the relevant aircraft accidents and collected all the available accident reports from 2006 to 2020. We chose this specific period for two main reasons. First, going too far back would mean having to include cases using technology that is incomparable with today's systems. Second, final reports require long investigations and most accidents that took place after 2020 are still under investigation.

As there is currently no single database that contains a detailed list of all aircraft accidents, including all the required information needed for the analysis of this study, the initial list of accidents was compiled using the information provided in Wikipedia [60]. To ensure the accuracy of information collected from this webpage, upon compiling the initial list, we verified the information used in the analysis by reviewing other websites such as The Aviation Herald (https://avherald.com, accessed on 1 September 2022), The Aviation Safety Network (https://aviation-safety.net, accessed on 1 September 2022), The National Transportation Safety Board (https://www.ntsb.gov, accessed on 1 September 2022), and several Civil Aviation Authority websites depending on the locations where the aircraft accidents took place (for example, the UK's aviation regulator (https://www.caa.co.uk, accessed on 1 September 2022)). It should be noted that depending on the locations where the accidents took place, the level of available information could vary significantly, depending, for example, on the depth of the investigation and the level of reporting provided by each authority.

Based on the identified accidents, those for which a final report was not available or not available in English were excluded. Any accidents that included helicopters, small private planes, utility planes, or gliders were also excluded from the study. Accidents that were attributed to hijacking were also excluded since their analysis was not relevant to the scope of this paper. In the next step, fewer selected reports were used to enable us to perform deeper analyses to understand the causes of these accidents and the specific actors involved (see Section 4.2). The reasoning behind choosing a smaller population of reports was three-fold. First, we wanted to focus on a smaller sample size for the sake of time, so that we were able to focus on understanding the specific causes of the accidents. Second, and more importantly, the first analyses on the full dataset showed that there was no statistical difference between focusing on the last 16 years or only the last 5 years of the reports. We carried out this statistical analysis by investigating the manufacturers, accidents, countries, and continents, and decided to focus on the last 5 years. Finally, we considered the changes in the aviation industry and decided that the older reports might not provide relevant information to be translated into the CPS domain because of the technological, cultural, and management-related changes in the industry.

### 4.2. Preliminary Analysis and Results

In total, 277 accidents were identified. For 102 of these accidents, the final report was available in English. Thirty-five reports were not in English, and the rest were either not available/public or not finalised. We decided to perform preliminary analyses on these 277 accidents by building a database in which commercial aircraft accidents were collected under the following categories: year, plane/helicopter, manufacturer, operator, country, departed from (city), departed from (country), departed from (continent), planned to arrive (city), planned to arrive (country), planned to arrive (continent), fatalities, reason, and

investigation results. The results of this analysis are presented later in this section, where we summarise the results of the analysis.

Table 1 shows the details of the accidents and fatalities divided by the continent where the accident happened. Most of the accidents and the fatalities took place in Asia. Although America had the second-highest number of accidents, Africa had the second-largest number of fatalities. Furthermore, 13.47% of the fatalities and approximately 10.83% of the accidents occurred in Europe. Another region with a relatively large number of accidents and fatalities was Eurasia, with 11.31% fatalities and 11.19% accidents. In the Mediterranean Sea, only one accident occurred, with 90 fatalities. Many accidents occurred in Oceania with smaller planes and fewer passengers, with 10 accidents and 74 fatalities in total. Australia and Antarctica had no accidents that fit the inclusion criteria of this study.

**Table 1.** Accident data categorised by continent, describing fatalities, number of accidents, and their relative percentages.

| Continent | Fatalities | % of Fatalities | Number of Accidents | % of Accidents |
|---|---|---|---|---|
| Asia | 3034 | 38.09% | 91 | 32.85% |
| Africa | 1623 | 20.38% | 45 | 16.25% |
| America | 1170 | 14.69% | 69 | 24.91% |
| Europe | 1073 | 13.47% | 30 | 10.83% |
| Eurasia | 901 | 11.31% | 31 | 11.19% |
| Mediterranean Sea | 90 | 1.13% | 1 | 0.36% |
| Oceania | 74 | 0.93% | 10 | 3.61% |
| Australia | 0 | 0.00% | 0 | 0.00% |
| Antarctica | 0 | 0.00% | 0 | 0.00% |

When we look at the statistics in more detail and focus on country (see Table 2), we can identify the countries where most of the accidents happened. For instance, the US was the site of most of the accidents in America, as was the case for Russia in Eurasia and the Democratic Republic of Congo in Africa. Even though Indonesia was the third country on the list, there was not a considerable difference between Indonesia, Nepal, or Iran. Most of the fatalities were shared between the following ten countries and regions in descending order: Russia, Indonesia, Iran, Brazil, Pakistan, Ukraine, the Indian Ocean, France, Democratic Republic of Congo, and Egypt. There was no obvious outliner in the European continent in terms of the number of accidents. Ukraine had four accidents and France and Italy had three. The rest of the accidents (20) were distributed between 18 different countries. However, most of the fatalities were shared between Ukraine, France, and Spain.

**Table 2.** Top ten countries with the most accidents.

| Country | Continent | Number of Accidents | Fatalities |
|---|---|---|---|
| US | America | 28 | 122 |
| Russia | Eurasia | 24 | 807 |
| Indonesia | Asia | 18 | 658 |
| Nepal | Asia | 13 | 212 |
| Democratic Republic of Congo | Africa | 13 | 333 |
| Iran | Asia | 12 | 598 |
| Pakistan | Asia | 7 | 501 |
| Brazil | America | 6 | 566 |
| Canada | America | 6 | 16 |
| Turkey | Eurasia | 5 | 60 |

We also looked at the airline companies and the numbers of accidents they had been involved in. Table 3 summarises the top ten airline companies represented in the data. Most

of the accidents (178 of 277) were single accidents and these were shared among 178 different airline companies. Repeated accidents occurred with highest rate with Southwest Airlines, Merpati Nusantara Airlines, Turkish Airlines, Asiana Airlines, EgyptAir, Ethiopian Airlines, FedEx Express, Iran Air, Pakistan International Airlines, and Qantas.

**Table 3.** Top ten airline companies with the most accidents.

| Airline | Country | Number of Accidents |
|---|---|---|
| Southwest Airlines | US | 6 |
| Merpati Nusantara Airlines | Indonesia | 4 |
| Turkish Airlines | Turkey | 4 |
| Asiana Airlines | South Korea | 3 |
| EgyptAir | Egypt | 3 |
| Ethiopian Airlines | Ethiopia | 3 |
| FedEx Express | US | 3 |
| Iran Air | Iran | 3 |
| Pakistan International Airlines | Pakistan | 3 |
| Qantas | Australia | 3 |

*4.3. Detailed Accident Report Analysis and Results*

As mentioned previously, we decided to focus our detailed analysis on accidents that took place between 2016 and 2020. Initially, 30 accidents were included in the detailed analysis. However, after completing the cause analysis of the reports, we identified that four of the reports were very brief and did not provide enough information. These four accidents were therefore also excluded from the NVivo® analysis. The details of the 30 accidents can be found in Table A1 in Appendix A.

The NVivo® analysis revealed 15 main contributing factors to accidents, arising from 722 references within the accident reports. Table 4 summarises these factors in descending order. It should be noted that there were cases in which a text reference could be attributed to more than one of the identified nodes. We describe these in more detail later in this section, including an example for each category.

**Table 4.** Contributing factors to accidents, as identified in the analysis.

| Contributing Factors | Number of Accidents | Number of References in Accident Reports |
|---|---|---|
| Non-adherence to SOPs [1] | 20 | 150 |
| Operational | 16 | 92 |
| Behaviours/Skills | 8 | 23 |
| Certifications | 6 | 8 |
| Maintenance/Service/Performance | 4 | 13 |
| Incident Reporting | 4 | 10 |
| General | 3 | 4 |
| Poor Handling | 20 | 139 |
| CRM | 15 | 52 |
| Emotional/Behavioural Issues | 12 | 30 |
| Equipment-Plane | 10 | 21 |
| Decision Making | 9 | 18 |
| Operational | 9 | 18 |
| Failure to identify safety issues | 19 | 119 |
| Lack of SOP | 17 | 58 |
| Lack of proper training | 15 | 42 |
| Lack of proper equipment | 12 | 28 |

**Table 4.** *Cont.*

| Contributing Factors | Number of Accidents | Number of References in Accident Reports |
|---|---|---|
| Lack of proper communication | 10 | 33 |
| Mechanical issue | 9 | 32 |
| Lack of safety culture | 9 | 28 |
| Poor weather conditions | 9 | 11 |
| Loss of situational awareness | 6 | 15 |
| Lack of proper maintenance | 5 | 22 |
| Lack of required personnel | 5 | 6 |
| Hierarchy issues | 4 | 7 |
| Design issue(s) | 3 | 11 |

[1] Standard operating procedures (SOPs).

- Non-adherence to SOPs: Despite the strict standard operating processes that exist in the aviation industry, in our study we identified several cases in which one or more of the actors involved decided to deviate from at least one standard operating procedure. This category was the most frequent contributing factor to accidents. This category refers to any of the actors involved in an accident, for example, the crew, ATC, the airline, or even the supervisory authority, etc., and includes cases in which a process was not followed in day-to-day operations, in certification processes, during maintenance, during incident reporting, etc. Some examples are mentioned below:

  *"The majority of the air operator's procedures, and in particular its safety management system, were only formal in nature and were not properly applied"*

  *"The flight crew was accustomed to not complying with recognised rules for safe flight operations and taking high risks"*

  *"The current regulation related to the personnel qualification for aerodrome personnel had not included several items as required by the ICAO standard, including requirement for radio telephony"*

  *"Numerous incidents, including several serious incidents, were not reported to the competent bodies and authorities. This meant that they were unable to take measures to improve safety"*

- Poor handling: This category refers to the poor handling of a situation by one or more of the actors involved in the accident. It can involve, for instance, poor decision-making, poor handling of equipment, poor crew resource management (CRM) on behalf of the crew, or even poor handling due to emotional or behavioural issues. Some examples are mentioned below:

  Emotional/Behavioural Issues: *"The first officer's long history of training performance difficulties and his tendency to respond impulsively and inappropriately when faced with an unexpected event during training scenarios at multiple employers suggest an inability to remain calm during stressful situations—a tendency that may have exacerbated his aptitude-related performance difficulties"*

  CRM: *"The flight crew did not effectively scan and monitor the primary flight instrumentation parameters during the landing and the attempted go-around"*

  Equipment: *"use of the automatic flight mode (autopilot, autothrottle) in the flight under the windshear conditions which resulted in the aircraft being unstable (excess thrust) when turning to the manual control"*

  Decision-making: *"The captain demonstrated inadequate aeronautical decision-making skills regarding which runway to use for landing and a lack of flight deck leadership by continuing the landing to a runway with a significant tailwind"*

Operational: *"Lack of effective oversight was observed in the part of operator as well as regulator at the departing airport"*

- Failure to identify safety issues: This category refers to cases in which at least one of the involved actors had the opportunity to identify a safety issue but failed to do so.

  *"Performance reviews were conducted by those responsible in a manner which lacked critical rigour and ignored errors"*

- Lack of SOP: This category refers to cases in which there were no proper SOPs available to one or more of the actors involved in the accident.

  *"In case of a missed approach, the OM* (Operations Manual) *- A Missed Approach flight procedures requires that the pilots shall advise air traffic control as soon as practicable. There was no guidance in the OM-A on what should be reported in case of a go-around"*

- Lack of proper training: This category refers to cases in which there were deficiencies in the training of one or more of the actors involved in the accident. This could include not just the flight crew, but also emergency services, ATC, etc.

  *"Experienced flight crews who often made mistakes regarding basic flying skills (airspace violations, non-compliance with basic rules) during flight operations showed deficits in terms of operation-specific training and collaboration"*

- Lack of proper equipment: This category refers to cases in which the proper equipment was not available at the time of the accident, contributing to the sequence of events. This could include equipment in the aircraft, the airport, ATC, training, etc.

  *"The absence of visual and aural alerts from both airplanes' traffic display systems, while operating in a geographic area with a high concentration of air tour activity"*

- Lack of proper communication: This category refers to cases in which there was a lack of proper communication between two or more of the actors involved in the incident.

  *"If the flight crew or the flight attendants had communicated after the airplane came to a stop, the flight crew could have become aware of the severity of the fire on the right side of the airplane and the need to expeditiously shut down the engines"*

- Mechanical issue: This category refers to cases in which a mechanical issue also contributed to the accident.

  *"Contributing to the accident were (1) Saab's design of the wheel speed transducer wire harnesses, which did not consider and protect against human error during maintenance"*

- Lack of safety culture: In this study, we adopted the definition proposed by Cox and Cox [51] for *"safety culture"*, defining it as the reflection of *"the attitudes, beliefs, perceptions, and values that employees share in relation to safety"*. This category therefore refers to cases in which one or more of the involved organisations or actors did not exhibit an appropriate culture that promoted a culture of safety.

  *"The flight crews who did not adhere to generally accepted principles for safe flying in mountainous areas when operating the type 'Ju 52/3m g4e' aircraft were often those who had trained as Air Force pilots. In particular, they systematically and significantly flew below safe altitudes and violated the minimum separation from obstacles"*

- Poor weather conditions: This category refers to cases in which poor weather conditions may have also contributed to the accident.

  *"An extreme loss of braking friction due to heavy rain and the water depth on the ungrooved runway, which resulted in viscous hydroplaning"*

- Loss of situational awareness: This category refers to any cases in which the flight crew had lost the ability to have an *"accurate perception and understanding of all the factors and conditions within the four fundamental risk elements (pilot, aircraft, environment, and type of operation) that affect safety before, during, and after the flight"* [61].

*"During the critical phase of final approach PIC (PF) lost situational awareness and deviated to the right by almost 15 degree and also descended below threshold height. He could not even pay attention to the FO's call out alerting the excessive descend"*

- Lack of proper maintenance: This category refers to cases in which any related equipment (including aircraft or airport equipment, emergency services equipment, etc.) may have contributed to the accident.

  *"In many instances, the quality of the remanufactured and reconditioned aircraft parts was poor."*

- Lack of required personnel: This category refers to cases in which a lack of required personnel contributed to the accident. This could be, for example, the case for the airline, the airport, the maintenance team, or even the supervisory authority.

  *"Though Lukla Tower was supposed to be two-man console, there was only one ATS Officer on duty at the time of accident."*

- Hierarchy issues: This category refers to cases in which an individual may have had to question a superior officer and therefore refrained from taking any action. Despite the strict hierarchical structures of authority that exist in the aerospace industry, this category only contributed to four of the analyzed accidents.

  *"Inability of the copilot (PM) to take control of the aircraft and proper action to execute go-around"*

- Design issue(s): This category refers to cases in which an aircraft design issue contributed to the accident; this was found to be relevant to only three accidents.

  *"Although the aircraft's ice-protection systems were activated on the approach to CZFD, the aircraft's de-icing boots were not designed to shed all of the ice that can accumulate, and the anti-icing systems did not prevent ice accumulation on unprotected surfaces. As a result, some residual ice began to accumulate on the aircraft."*

Our analysis also revealed the different actors that were involved in the analyzed accidents. Table 5 summarises these actors in descending order. It should be noted here that the identified actors may not have necessarily caused the accident but rather that they affected the sequence of events and therefore the final outcome of the accident. For example, in the case of passengers, it could mean that they did not follow the crew's instructions when evacuating the aircraft, leading to a higher number of injuries. In one of the analyzed accidents, although it was not clear in the report exactly which actors were involved, it was clear that the accident was due to multiple actors:

*"The accident was caused by insufficient operational prerequisites for the management of a failure in a redundant system"*

**Table 5.** Different actors involved in the analyzed aircraft accidents.

| Actors | Number of Accidents | Number of References in Accident Reports |
|---|---|---|
| Flight Crew | 21 | 198 |
| Airline | 18 | 137 |
| Supervisory Authority | 15 | 67 |
| Air Traffic Control (ATC) | 12 | 40 |
| Aircraft Manufacturer | 8 | 43 |
| Airport | 5 | 22 |
| Maintenance Team | 3 | 23 |
| Emergency Services | 3 | 13 |
| Passengers | 2 | 3 |
| Airport Crew | 2 | 4 |
| Flight Crew (previous flight) | 1 | 3 |
| Multiple Actors | 1 | 2 |

What is interesting to note is that when looking at all the analyzed accidents more closely, most of the accidents were the result of multiple actors and multiple contributing factors. Only one accident was attributed solely to a mechanical issue without identifying any clear actors in the report. In three accidents there were at least two contributing factors and only one actor identified in the report, whereas for the rest of the analyzed accidents there were at least two actors and two or more contributing factors. In the most complex accident, we identified 13 contributing factors and 8 different actors involved. Table 6 presents the numbers of actors and contributing factors involved per accident.

**Table 6.** Numbers of actors and contributing factors involved in the analyzed aircraft accidents.

| Accident Code | Actors | Contributing Factors |
| :---: | :---: | :---: |
| N109 | 8 | 13 |
| N72 | 8 | 13 |
| N59 | 7 | 8 |
| N87 | 6 | 12 |
| N61 | 5 | 13 |
| N75 | 5 | 9 |
| N104 | 4 | 14 |
| N107 | 4 | 9 |
| N123 | 4 | 8 |
| N128 | 4 | 10 |
| N79 | 4 | 7 |
| N112 | 3 | 7 |
| N133 | 3 | 13 |
| N66 | 3 | 5 |
| N78 | 3 | 10 |
| N82 | 3 | 12 |
| N91 | 3 | 11 |
| N97 | 3 | 12 |
| N64 | 2 | 10 |
| N116 | 2 | 4 |
| N122 | 2 | 2 |
| N86 | 2 | 6 |
| N118 | 1 | 4 |
| N119 | 1 | 3 |
| N98 | 1 | 2 |
| N88 | 0 | 1 |

## 5. Discussion

The main scope of this study was to understand the safety and autonomy practices of the aviation industry by reviewing accident reports and to transfer these findings to the design and implementation of autonomous CPSs. In this section, we have attempted to provide a few guidelines to help transfer these findings.

The findings showed how difficult it is to identify a specific actor or single cause for the accidents. This difficulty arose from the inherited complexity of the events. The accidents rarely occurred because of one single cause or actor, but were generally the result of several cascading events. Therefore, on the one hand, we tried to approach the accidents from a systematic perspective to categorise "actors and causes" in order to understand the accidents. On the other hand, our aim was not to perform a comprehensive statistical analysis in order to identify the most important causes or the most influential actors. Instead, we focused on transferring all the findings obtained via the analyses of these accidents into the design and operation of a CPS in order to increase the knowledge in this domain by translating the information learned in the analysis. One of the first findings obtained in this effort to transfer the results of this study concerned the exact point mentioned above—accidents are not easy to understand, and it can be difficult to pinpoint only one cause or actor because of the nature of cascading events and failures.

Autonomous systems in general and autonomous CPSs are products which include many actors. They perform critical actions relating to safety and in the case of accidents it is certain that identifying a cause and one direct link to an actor will not be possible. Therefore, in the CPS domain, industry, researchers, and relevant stakeholders should consider developing mechanisms to make this identification process easier. Therefore, to deal with the high complexity of accidents, the first guideline we propose is to consider the roles and responsibilities of different actors and make them clear and linkable to different outcomes (causes of accidents).

The main research question of this article was *"What are the important considerations in the design of future autonomous and intelligent systems (CPSs) when focusing on the results regarding the safety of decision-making processes and the accountability of these decisions?"*

The findings of the study showed that we need to think about the actors, causes, and possible factors which may contribute to safety-related issues from the very beginning of the CPS design and implementation process. The current practices focused on building functioning CPSs and current safety practices are not sufficiently well defined to deal with the human-in-the-loop decision-making process. In particular, the accountability of different stakeholders, as well as the cascading effects of accountabilities and failures and their effects on different groups of people, including the distribution of accountability in safety practices, should be considered in the design of the next generation of CPSs. To this end, in order to deal with safety-related issues, the second guideline we propose is to acquire a safety mindset and implement safety practices throughout the product development process from the design phase up to the operation and maintenance of the CPS.

Moreover, there should be well-defined procedures and supporting processes to identify accountability when safety-related issues arise in CPSs. We found accident reports to be beneficial when we tried to understand the roles, causes, and consequences of the accidents in this study. However, these reports were compiled by different institutions, in different time periods, and with different structures and layouts. This diversity in the reporting of accidents made the analysis difficult. There is no well-defined procedure or official database for reports at this moment but there is certainly a need to employ one, not only for the aviation industry but also for other relevant domains such as CPSs. Therefore, as our third guideline, we suggest starting an immediate domain-level effort to design and distribute accident reporting practices, templates, and similar resources, in which links between roles, causes, and consequences can be easily established, thus making it possible to analyse these reports in a more systematic and analytical manner to improve the benefits of accident reports and reporting practices.

To examine these findings in greater detail, we also asked the following research sub-question: *"What can we learn from the aviation industry in regard to human-in-the-loop decision-making by reviewing accident reports and considering accountability and safety as the main two metrics?"*

We found that, in most accidents, several actors (rather than just one) were involved. Several contributing factors were also behind the causes of the accidents. It was hard to separate one single event from the many that led to each accident. Even in the reports, it was not always easy to separate the actual causes from the contributing factors. A definitive method to approach the fundamental challenge of identifying the cause of a single event is not available even in the aviation industry, where incredible efforts have been made to standardise the investigation and reporting processes over several decades. The CPS industry can learn from this, as newly designed CPSs such as autonomous vehicles currently lack safety standards, procedures, or sufficient datasets pertaining to accidents. Furthermore, these CPSs do not have established accident-reporting practices or official mechanisms which can be used to learn from these data. The CPS industry should initiate discussions regarding storing and allowing access to data pertaining to accident reports, in addition to providing systematic ways to collect these data.

Another important finding was related to standard operating procedures (SOPs). Our analysis revealed that the most frequent cause of accidents was "non-adherence to SOPs". Considering that the aviation industry has had the opportunity to establish and improve these over the course of several decades, it can be observed that having an SOP in place does not necessarily mean that all safety issues will be solved. The CPS industry can learn from this and focus not only on providing a standardised approach, but also on including different levels of mechanisms, which can be checked by diverse stakeholders, in order to minimise the cascading nature of such events and to be able to address stakeholder accountability and improve safety. Therefore, we suggest that the industry should not focus solely on the standardisation of operational procedures because, while this is an effective way to improve safety at the beginning, it is not a solution for all potential issues.

Finally, our second research sub-question was "*How can these findings then be translated to developing a methodological approach to be used in design of autonomous CPSs?*"

In our analysis, we identified a list of actors that either directly or indirectly led to an accident through their actions, noting the actors that were found to be involved in most of the accidents. Even though the actors whose actions may lead to accidents in a CPS system may differ from those in the aviation industry, most of them can be directly linked to a relevant actor identified in our analysis. For example, the actor that was involved in most of the accidents was the "flight crew". Therefore, the CPS industry can investigate the potential actions of the operating crews of their systems which may lead to accidents. The next two most frequently involved actors were the operator (the airline) and the supervisory authority (e.g., the Federal Aviation Administration); both of these can similarly be mapped to relevant actors in a CPS. Although the easiest actor to identify was often the operating crew, upon further analysis, supervisory authorities' failures to notice missing links between procedures and accountability were the real causes of many events. In highly autonomous next-generation CPSs, it is likely that similar authorities will be needed, and their roles should be defined in such a way as to ensure that these mistakes are not repeated.

We also analysed the most common actions that could lead to accidents. For example, the most common factor that led to an accident in our sample was the failure by one or more of the involved actors to adhere to an SOP. As mentioned previously, the CPS industry can learn from this and focus on promoting accountability in order to improve overall safety. Another common issue identified was the failure by one or more of the actors involved to identify a safety issue. Indeed, during our review of the relevant literature, we identified some key issues related to the wider understanding of safety. First, all available definitions of safety presented to date are relative and related to the way in which one perceives risk. Second, several of the foundations and assumptions of the relevant literature were found to be based on what Hollnagel et al. [45] defined as safety, thus oversimplifying systems and treating the human element as a liability. However, in today's systems, everything is connected and very few components of these systems are independent. Hence, solutions must be based on socio-technical rather than technological solutions. Furthermore, processes are not complete, and even when people properly understood these processes, they do not always behave in the way that they are supposed to. Providing too many rules can take away personal accountability, as it makes it easier to pass the blame onto others [62]. To find a solution that works in today's conditions of complexity, more efforts need to be made in order for organisations to recognise humans as an asset rather than a liability [45], especially in CPSs. To this end, we suggest promoting accountability, providing targeted training to identify safety issues, providing motivation, and implementing feedback mechanisms to improve procedures, hence enabling the development of socio-technical solutions.

In addition, according to the definition of TLA adopted here, organisations have to understand the boundaries and responsibilities needed to offer safe solutions, in order to inform, justify, and accept the consequences of decisions and actions taken. However, is TLA all about assigning blame and justifying one's actions, or is it more about understanding the

consequences before making decisions and taking actions in order to reduce undesirable consequences? Senders and Moray [63], for instance, suggest that it is more important to reduce the consequences of wrong actions, rather than focusing on the actual actions, in order for the organisation to benefit from trial and error and increase flexibility. This highlights the need for a better understanding of Safety II and for a more pre-emptive definition of TLA, one that can help organisations evolve and improve. Based on the issues discussed above, we propose the following definition of safety:

> *"Safety is the process of constantly driving towards zero incidents by managing both expected and unexpected hazards."*

A recurring issue identified in our review of the literature on accountability was that accountability has a complex meaning and multiple dimensions that can lead to conflicting situations, since actors are often required to give account to a variety of different stakeholders against a broad range of criteria [40]. Although in our analysis we were able to identify the different actors that had control over the sequence of events during an aircraft accident, there was no clear evidence of conflicting accountabilities. To be more specific, it did not seem that actors were considering their actions according to who might call them to give account. Put differently, the analysis did not show actors to be involved in any kind of "internal struggle" with respect to which "authority" would call them to account for the consequences of their actions. Organisations that operate in sectors where safety is a key consideration, such as organisations operating in CPSs, need to operate as high-reliability organisations. According to Sullivan and Beach [64], for example, members of high-reliability organisations need to be aware of how their performance affects the performance of the organisation as a whole through "visibility" and "accountability". This does not seem to be the case for the different actors involved in aircraft accidents.

The analysis further identified a human redundancy issue. This was the case when one of the actors felt that someone else was responsible for doing the job and therefore failed to take proper actions, with potentially catastrophic consequences. Downer [65] explained, for example, that "*two engines on an aircraft are safer than four*", implying that more does not necessarily mean better. An accident that highlights this observation is that involving a UTair flight in Russia in 2012, which was caused due to the buildup of ice on the wings of the aircraft. According to standard procedures, although the ground staff was responsible for de-icing the aircraft, the final responsibility for accepting that the plane was in a state to fly was the pilot's. According to post-accident interviews, the ground staff stated that the wings' surface had no ice. However, the investigators pointed out that the ground staff did not use a ladder to inspect the wings and therefore their job had not been performed properly. When the ground staff gave the go-ahead, the pilot departed without performing any further checks. As a result of the ice, the plane crashed, and 33 lives were lost. The human redundancy here lay in the failure of the ground staff to perform the necessary procedures on the basis that the final decision was the pilot's responsibility. Therefore, the analysis highlighted that a balance needs to be found between human redundancy and systems redundancy, as this might improve the safety of the system. As our final guideline, we suggest that the CPS industry should consider various principles in order to operate as a high-reliability organisation, such as deference to expertise, reluctance to simplify, sensitivity to operations, a commitment to resilience, and a preoccupation with failure [66], in order to balance between human and systems redundancy.

## 6. Conclusions

In this study, we first identified 277 accidents in total. The first analyses showed that 33% of the accidents and 38% of fatalities happened in the continent of Asia and in the country of Indonesia (18 accidents and 658 fatalities). We then analyzed 26 aircraft accidents from 2016 to 2022 in order to understand how autonomy and safety was managed in the aviation industry and to transfer our findings to the study of CPSs. We found that the complex meaning of safety and the complex nature of accidents mean that acquiring an understanding of safety is far from straightforward. Interestingly we found that to

date, there is no commonly accepted definition of safety. Furthermore, we found that most accidents were not the result of a single actor or factor but were rather the result of a sequence of events and several different actors. We also found that despite the strict and long-standing SOPs that existed in the aviation sector, the most frequent factor identified was "non-adherence to SOPs". We therefore make the following recommendations.

- It is necessary to consider the actors, causes, and possible factors which may contribute to safety-related issues from the very beginning of the CPS design and implementation processes. The list of actors and factors identified in this study can form the basis for further analyses towards this goal in relation to CPSs.
- Although there should be well-defined procedures and supporting processes to identify accountability when safety-related issues arise in CPSs, there is also a need for both well-defined reporting procedures/practices and different levels of mechanisms that can be checked by diverse stakeholders in order to minimise the cascading nature of events and to be able to address stakeholder accountability and improve safety. Further research into high-reliability organisations (HROs) [67] can provide further insights for CPS. The HRO literature emerged following a number of tragic accidents, such as NASA's Challenger explosion [67], when a new group of scholars proposed the HRO paradigm and focused their research on organisations that could not afford to fail [68–70]. A relevant point regarding the HRO literature is that its focus has shifted from identifying how accidents happen to identifying successful organisations [67]. HRO scholars have suggested that successful organisations that operate in hazardous conditions require the following characteristics: a culture of continuous improvement and learning, flexible structures, commitment to results and safety, a culture of reporting failures, effective communication, in-built human and system redundancy, outstanding technology, a commitment to standard operating procedures, and the establishment of minimum requirements [64,67,68,70–79]. The attributes identified in this study are clearly relevant to further research in this area, especially in regard to finding the right balance between flexible structures and standard procedures, as well as reporting incidents can provide further insights to ensure the safety of CPSs.
- We also propose the following definition of safety: "*Safety is the process of constantly driving towards zero incidents by managing both expected and unexpected hazards.*"

Like any other scientific study, this study has limitations that we would like to mention before concluding the article. We outlined several inclusion and exclusion criteria in the methodology section which enabled us to narrow down the selected reports and to conduct detailed analyses. We only read reports which were published in English, and some accident reports related to more recent accidents were not ready when this article was written. Even though we tried our best to minimise bias and provide high-quality research findings, only two of the team's researchers read the reports and gathered the findings.

**Supplementary Materials:** The initial list of aircraft accidents, including a summary of each one, can be found at https://en.wikipedia.org/wiki/List_of_accidents_and_incidents_involving_commercial_aircraft#2006 accessed on 1 March 2022.

**Author Contributions:** Conceptualization, C.M.; Data curation, C.M. and D.G.B.; Formal analysis, C.M.; Methodology, C.M., D.G.B. and A.N.; Supervision, A.N.; Visualization, D.G.B.; Writing—original draft, C.M. and D.G.B.; Writing—review & editing, C.M. and D.G.B. All authors have read and agreed to the published version of the manuscript.

**Funding:** This research received funding from: i. the Centre for Digital Built Britain's (CDBB) at the University of Cambridge, which is within the Construction Innovation Hub and is funded by UK Research and Innovation through the Industrial Strategy Fund and the European Union's Horizon 2020 research, UKRI grant No. 104513 and ii. the innovation Programme under the Marie Skłodowska-Curie grant, agreement No. 882550.

**Institutional Review Board Statement:** Not applicable.

**Informed Consent Statement:** Not applicable.

**Data Availability Statement:** Not applicable.

**Acknowledgments:** The first two authors of this article are grateful to their daughters, Ioanna Daskalaki and Idun Aishe Broo, for timing their births perfectly so that their mothers had enough time to work on, write up, and revise this work smoothly.

**Conflicts of Interest:** The authors declare no conflict of interest.

## Appendix A

**Table A1.** List of accidents.

| Accident Code | Year | Manufacturer | Operator | From (Country) | Into (Country) | Fatalities |
|---|---|---|---|---|---|---|
| N59 | 2016 | Bombardier | West Air Sweden | Norway | Sweden | 2 |
| N61 | 2016 | DHC | Tara Air | Nepal | Nepal | 23 |
| N64 | 2016 | Boeing | Flydubai | UAE | Russia | 62 |
| N66 | 2016 | Boeing | Batik Air | Indonesia | Indonesia | 0 |
| N72 | 2016 | Boeing | Emirates | India | UAE | 1 |
| N75 | 2016 | Boeing | American Airlines | US | US | 0 |
| N78 | 2016 | BAE | LaMia Airlines | Bolivia | Colombia | 71 |
| N79 | 2016 | ATR | Pakistan International Airlines | Pakistan | Pakistan | 47 |
| N82 | 2017 | Boeing | Turkish Airlines | Kyrgyzstan | Kyrgyzstan | 4 |
| N86 | 2017 | Let Kunovice | Summit air | Nepal | Nepal | 2 |
| N87 | 2017 | Airbus | Air Canada | Canada | US | 0 |
| N88 | 2017 | Airbus | Air France | France | Canada | 0 |
| N91 | 2017 | ATR | West Wind Aviation | Canada | Canada | 1 |
| N97 | 2018 | Bombardier | US-Bangla Airlines | Bangladesh | Nepal | 51 |
| N98 | 2018 | Boeing | Southwest Airlines | US | US | 1 |
| N104 | 2018 | Junkers | Horizon Air | Switzerland | Switzerland | 20 |
| N107 | 2018 | Boeing | Utair | Russia | Russia | 0 |
| N109 | 2018 | Boeing | Lion Air | Indonesia | Indonesia | 189 |
| N112 | 2019 | Boeing | Atlas Air | US | US | 3 |
| N116 | 2019 | Boeing | Miami Air | Cuba | US | 0 |
| N118 | 2019 | Bombardier | Biman Bangladesh Airlines | Bangladesh | Bangladesh | 0 |
| N119 | 2019 | DHC | Mountain Air Service | US | US | 1 |
| N122 | 2019 | Antonov State Enterprise | Ukraine Air | Spain | Ukraine | 5 |
| N123 | 2019 | Saab | PenAir | US | US | 1 |
| N128 | 2020 | McDonnell Douglas | Caspian Airlines | Iran | Iran | 0 |
| N133 | 2020 | Boeing | Air India Express | UAE | India | 20 |

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
