# Peer review of "Human-in-Loop Decision-Making and Autonomy: Lessons Learnt from the Aviation Industry Transferred to Cyber-Physical Systems"

_technologies, doi:10.3390/technologies10060120_

Round 1

Reviewer 1 Report

Paper deals with important task. The authors reviewed aircraft accidents in order to understand how autonomy and safety is managed in aviation industry and transfer the learnings to autonomous Cyber Physical Systems.

Paper has great practical value.

Suggestions:

1.       The abstract section should be extended using more clearly the motivation of this paper.

2.       The methodology section is unclear. If you used databases like Scopus, WoS please extend this section using for example PRISMA scheme. If you used some historical data please argue his choice

3.       It is unclear how the authors have identified the relevant aircraft accidents and collected all the available accident reports from 2006 to 2020 What database used for that and why

4.       The conclusion section should be extended using: 1) numerical results obtained in the paper; 2) limitations of the conducted research; 3) prospects for future research.

5. Most of all references are outdated. Please fix it using 3-5 years old papers in high-impact 

Reviewer 2 Report

The manuscript proposes an interesting analysis of accidents occurred in aviation industry, with the purpose of transferring the lessons learnt to the world of cyber-physical systems. I find the general idea of transferring the experience of aviation to cyber-physical systems interesting and worthwhile being investigated, but the discussion on how to transfer the lessons learnt to the domain of interest (cyber-physical system) is quite limited and it provides only a list of generic and not really concrete proposals. I would have expected the proposal of a set of more concrete guidelines to be applied to the cyber-physical systems domain. Most of the manuscript is dedicated to analyzing accidents occurred in the aviation industry, but then details are lacking in the part that is of most interest to the potential readers of this journal.

Additional comments follow:

- In the Abstract: the meaning of the last sentence of this section ("The review of High-Reliability Organisations can offer further insights in future research of
CPS safety.") is not clear to me.

- There are sections (e.g., Section 2) where subsections start right away without any introduction (i.e., Section 2.1 starts right after the tile of Section 2).

- In the different tables where locations (e.g., the "Continent" column in Table I) are mentioned, where it is not really clear if departure or arrival (or both) locations are considered. This should be stated clearly.

- The NVivo software application is mentioned multiple times without any explanation. For the paper to be self-contained, a short introduction (few sentence) on this software application should be provided.

Round 2

Reviewer 1 Report

Paper can be accepted

Author Response

Thank you very much for all the comments and feedback.

Reviewer 2 Report

The authors satisfactory addressed all my "additional comments" and the comments of the other reviewer, but they completely disregarded my main concern, that I recall from my previous review:

I find the general idea of transferring the experience of aviation to cyber-physical systems interesting and worthwhile being investigated, but the discussion on how to transfer the lessons learnt to the domain of interest (cyber-physical system) is quite limited and it provides only a list of generic and not really concrete proposals. I would have expected the proposal of a set of more concrete guidelines to be applied to the cyber-physical systems domain. Most of the manuscript is dedicated to analyzing accidents occurred in the aviation industry, but then details are lacking in the part that is of most interest to the potential readers of this journal.

This comment was the main reason for suggesting a major revision and, unfortunately, this reason still stands.

Author Response

Please find attached file for explanations.

Round 3

Reviewer 2 Report

The authors modified the paper to include all the suggested modifications in a satisfactory way.

The only minor comment that remains is related to the bullet of line 738: is it a "sub-bullet" of the bullet of line 734 (which does not make much sens, being only one) or is it just an indentation error?